# Selective Intra-Arterial Doxorubicin Eluting Microsphere Embolization for Desmoid Fibromatosis: A Combined Prospective and Retrospective Study

**DOI:** 10.3390/cancers14205045

**Published:** 2022-10-14

**Authors:** Eldad Elnekave, Eytan Ben Ami, Sivan Shamai, Idit Peretz, Shlomit Tamir, Elchanan Bruckheimer, Amos Stemmer, Joseph Erinjeri, Abed Abu Quider, Max Seidensticker, Moritz Wildgruber, Jens Ricke, Antoinette Anazodo, Kin Fen Fung, Alona Zer, Shifra Ash

**Affiliations:** 1Unit of Interventional Radiology, Shaare Tzedek Medical Center, Jerusalem 9103102, Israel; 2Department of Diagnostic Radiology, Rabin Medical Center, Petah Tikva 4941492, Israel; 3Sarcoma and Bone Oncology Unit, Oncology Division, Chaim Sheba Medical Center, Ramat Gan 52621, Israel; 4Sarcoma and Bone Oncology Unit, Oncology Division, Tel Aviv Sourasky Medical Center, Tel Aviv 6423906, Israel; 5Davidoff Cancer Institute, Rabin Medical Center, Petah Tikva 4941492, Israel; 6Section of Pediatric Cardiology, Schneider Children’s Medical Center, Petah Tikva 4941492, Israel; 7Oncology Division, Chaim Sheba Medical Center, Ramat Gan 52621, Israel; 8Interventional Radiology Service, Memorial Sloan Kettering Cancer Center, New York, NY 10065, USA; 9Department of Pediatric Hematology Oncology, Saban Pediatric Medical Center, Soroka University Medical Center, Beer Sheva 84101, Israel; 10Department of Radiology, University Hospital, Ludwig Maximilian University, 80539 Munich, Germany; 11Kids Cancer Centre, Sydney Children’s Hospital, Randwick, Sydney, NSW 2031, Australia; 12Department of Radiology, Hong Kong Children’s Hospital, Hong Kong 999077, China; 13Department of Oncology, Rambam Medical Center, Haifa 3109601, Israel; 14Department of Pediatric Hematology-Oncology, Rambam Medical Center, Haifa 3109601, Israel

**Keywords:** angiography, desmoid fibromatoses, doxorubicin eluting embolization (DEE), chemoembolization

## Abstract

**Simple Summary:**

Desmoid fibromatoses (DFs) are locally aggressive tumors composed of monoclonal fibroblasts within an abundant extracellular matrix. Systemic treatment with doxorubicin is effective, but associated with significant toxicity. We investigated arterial doxorubicin eluting embolization (DEE), an approach that delivers high doxorubicin concentrations to the tumor with limited systemic drug exposure, in 24 patients (median age, 24 years; interquartile range, 16–34). Most patients (71%) had one or more than one prior DFs treatment (surgery, systemic therapy, or both). Patients underwent a median of two (range, 1–4) DEE treatments, with a median of 49 mg (range, 8–75) doxorubicin per treatment. Efficacy outcomes were available for 23 patients. With a median follow-up of 8 months (interquartile range, 3–13), median tumor volumes decreased by 59% (interquartile range, 40–71%). Of 23 patients, 9 (39%), 12 (52%), and 2 (9%) had a partial response, stable disease, and progressive disease, respectively. The procedure was safe and well tolerated.

**Abstract:**

Desmoid fibromatoses (DFs) are locally aggressive tumors composed of monoclonal fibroblasts within an abundant extracellular matrix. Systemic doxorubicin treatment is effective, but toxic. We investigated arterial doxorubicin eluting embolization (DEE), an approach characterized by high drug concentrations in the tumor alongside limited systemic drug exposure. The primary and secondary endpoints were radiological response using MRI and RECIST 1.1, respectively. The study included 24 patients (median age, 24; interquartile range, 16–34 years). Data were collected prospectively for 9 patients and retrospectively for 15 patients. The most frequent tumor locations were chest/abdomen wall and neck/shoulder/axilla (29% each). Of 24 patients, 7 (24%) were treatment naïve, and 17 (71%) had received one or two prior treatments. Patients underwent a median of two treatments (range, 1–4), with a median of 49 mg (range, 8–75) doxorubicin/treatment. Efficacy outcomes were available for 23 patients. With a median follow-up of 8 months (interquartile range, 3–13), median tumor volumes decreased by 59% (interquartile range, 40–71%) and T2 signal intensity decreased by 36% (interquartile range, 19–55%). Of 23 patients, 9 (39%), 12 (52%), and 2 (9%) had a partial response, stable disease, and progressive disease, respectively. DEE was safe and well tolerated, with one reported grade 3–4 adverse event (cord injury). In conclusion, DEE was safe and achieved rapid clinical/volumetric responses in DFs.

## 1. Introduction

Desmoid fibromatoses (DFs) are rare (1–2 cases/million yearly), locally aggressive mesenchymal tumors, characterized histologically by monoclonal myofibroblasts within abundant stromal tissue. Most DFs cases (>85%) arise sporadically and the rest (5–15%) are associated with familial adenomatous polyposis (FAP) [1,2]. Nearly a quarter of asymptomatic DFs regress spontaneously, whereas the remainder progress along a variable course of growth and invasion into adjacent neurovascular structures and viscera [3]. Patients commonly experience chronic pain, impaired physical function, and insomnia as well as restricted social and professional engagement [4]. Up to 24% of patients with intra-abdominal DFs die of the disease [5].

Contemporary management has moved away from primary resection [6,7,8,9,10], as recurrences after resection are frequent and their phenotype is often more infiltrative [11,12,13,14,15,16,17]. Non-surgical approaches remain sub-optimal. For asymptomatic disease, contemporary guidelines recommend an initial interval of active surveillance [18]. Using this approach, up to half of all patients do not require treatment, with a median follow-up of 5 years [19]. For progressive or symptomatic disease, the benefit derived from systemic therapies must be weighed against their toxicity profiles. Doxorubicin is routinely used to treat soft-tissue sarcomas and other mesenchymal malignancies and is efficacious in DFs [20,21,22]; however, its use is associated with hematological, gastrointestinal, and cardiac toxicities [23,24,25]. Therefore, doxorubicin is generally reserved for non-responsive, symptomatic, rapidly growing and/or life-threatening DFs [20,22,26].

We hypothesized that the intrinsic hypervascularity of DFs tissue could be leveraged as a conduit to achieve local delivery of doxorubicin via endovascular catheter navigation. Doxorubicin contains a protonated amine group which can be ionically bound to sulfonate-coated hydrogel microbeads [27], allowing for microbead-loaded doxorubicin eluting embolization (DEE). DEE achieves sustained, high doxorubicin concentrations in the target tissue [28,29] with low systemic plasma concentrations [30,31]. 

We previously reported regarding a proof-of-concept study investigating DEE for extra-abdominal DFs in four children [32,33]. Here, we report regarding the efficacy and safety of DEE for intra- and extra-abdominal DFs in 24 patients treated with 52 DEE sessions in six tertiary medical centers.

## 2. Materials and Methods

### 2.1. Study Design and Patients

This report combines data collected prospectively and retrospectively. For both, eligible patients had to have histological confirmation of DFs with a long diameter ≥ 30 mm in an anatomical location accessible for endovascular treatment, and magnetic resonance imaging (MRI) evidence of T2 hyperintensity. Exclusion criteria included concurrent participation in another interventional study, uncontrolled intercurrent illness such as an active infection or symptomatic congestive heart failure (New York Heart Association Class III or IV), treatment with anthracyclines at cumulative doses ≥ 360 mg/m^2^, or a history of allergic reaction attributed to doxorubicin. 

Institutional review board (IRB) approval for the prospective trial (ClinicalTrials.gov number: NCT03966742) was obtained at Rabin Medical Center (RMC) and approval for retrospective data analysis was obtained from each additional participating institution. All patients provided written informed consent prior to treatment. 

### 2.2. Treatment 

All patients underwent super-selective arterial catheterization using 4 Fr or 5 Fr guiding catheters and 2.7 Fr–1.6 Fr coaxial micro-catheters. Diagnostic angiography was performed immediately prior to each treatment. Tumor vessels were identified based on anatomic origin, morphologic features (hypertrophy and/or abnormal tortuosity), and flow into the angiographic “tumor blush” of the DFs. 

DEE was performed using doxorubicin eluting microparticles. One vial containing 2 mL of 75–150 μM DC Beads (Boston Scientific, Ltd Marlborough, MA, USA.) or 2 mL 100 μM LifePearls (Terumo, Ltd, Shibuya City, Tokyo, Japan.) was loaded with 75 mg doxorubicin per treatment. Each vial was then diluted to 10 mL using 4 mL saline and 4 mL Omnipaque iodinated contrast material. The embolization endpoint was delivery of the doxorubicin dose using the minimal amount of embolic material, stopping short of arterial stasis if the vascular distribution was insufficiently large to receive the entire dose. Consolidation of blood flow toward the tumor and away from non-target tissue was achieved using standard angiographic techniques such as micro-coil embolization of small arteries (e.g., intercostal, internal mammary, and inferior epigastric) and temporary balloon occlusion of large ones (e.g., brachial) distal to the tumor. Intra-procedural cone beam computed tomography (CT) was used as needed to confirm tumor coverage and ensure the exclusion of flow to non-target tissue. 

### 2.3. Endpoints and Assessments

The primary endpoint was radiological response. MRI scans were obtained at baseline, 2–4 months after each treatment, at 6-month intervals following completion of treatment for the first 2 years, and annually thereafter. Characteristic MRI features of DFs include a heterogeneously hyperintense T2 or a short TI inversion recovery (STIR) signal among interspersed hypo-intense bands [34]. Histologically, the ratio between a T2 signal versus hypo-intense bands correlates to the degree of cellularity versus fibrotic matrix within DFs [35,36,37]. Decreased tumor volume and loss of T2 intensity are reliable markers of DFs responses to systemic treatment [38,39] and served as primary imaging metrics in the current study. Semi-quantitative T2 signal intensity was calculated by comparing the quantitative MRI signal within the tumor to that of an adjacent muscle, which served as an internal control. In addition to tumor volume and loss of T2 intensity, the greatest diameter of the tumor was also evaluated at each follow-up MRI. 

In addition to the MRI response, the revised Response Evaluation Criteria in Solid Tumors (RECIST 1.1) [40] was used as another primary imaging endpoint. 

### 2.4. Statistical Analysis 

Descriptive statistics were used to summarize the results. Statistical analysis was performed using R version 4.0 (www.r-project.org, accessed on 24 May 2021) [41].

## 3. Results

### 3.1. Study Patients and Treatments Received

The analysis included 24 patients treated in 52 sessions between March 2014 and August 2021. Nine patients participated in the prospective study at RMC (Israel) in 2019–2020. The remaining 15 patients included in the retrospective multicenter study were treated in Schneider Children’s Medical Center (Israel, 2014–2018; *n* = 7), Memorial Sloan Kettering Cancer Center (US, 2020; *n* = 4), Ludwig Maximilian University Hospital (Germany, 2020; *n* = 1), Hong Kong Children’s Hospital (China, 2020; *n* = 1), and Sydney Children’s Hospital (Australia, 2020; *n* = 1). 

Patient demographics, tumor locations, prior treatments, and DEE treatments are summarized in Table 1. The median age at time of treatment was 24 years (interquartile range (IQR), 16–34). The most frequent tumor locations were chest/abdomen wall and neck/shoulder/axilla (29% each), and extremities (25%). Twenty tumors (83%) were extra-abdominal and superficial and 4 (17%) were deep, including two intra mesenteric tumors, one pelvic tumor and one posterior mediastinal tumor. The median tumor volume at baseline was 310 mL (IQR, 108-686) and the median largest dimension was 10.5 cm (IQR, 9.35–13.925). Of 24 patients, 7 (24%) were treatment naïve (refused prior treatments), 9 (38%) received one prior treatment modality (typically surgery or systemic therapy), and 8 (33%) received two treatment modalities (e.g., surgery and systemic therapy). All patients had progressive or symptomatic disease at the time of DEE. 

The median time from diagnosis to DEE was 2.5 years (IQR, 2.0–3.6). Each patient underwent a median of two treatments (range, 1–4), interspersed 2–4 months apart. The median total delivered doxorubicin dose was 75 mg (range, 8–269) (median per treatment, 49 mg (range, 8–75)).

### 3.2. Efficacy

Efficacy outcomes were available for 23 patients (one experienced an intraprocedural vascular injury and did not complete treatment). With a median follow-up of eight months (IQR, 3–13), median tumor volumes decreased by 59% (IQR, 40–71%) and T2 signal intensity decreased by 36% (IQR, 19–55%). Of 23 patients, 9 (39%), 12 (52%) and 2 (9%) experienced a partial response (PR), stable disease (SD), and progressive disease (PD), respectively (Table 2). A representative image showing decreased T2 signal intensity is presented in Figure 1. Efficacy analysis was also stratified by duration of follow-up, as 7 patients (30%) had follow-up of at least a year, whereas 16 patients (70%) had a shorter follow-up (Table 2 and Figure 2). Patients in the longer follow-up group, who also had more DEE procedures, had better clinical outcomes compared to patients in the shorter follow-up group (Table 2).

### 3.3. Safety

Mean inpatient stay following the procedure was 1.2 days (range, 0–7). Of 52 procedures, 44 (85%) were performed as either outpatients or with a single night observation. During observation, three patients experienced transient leukocytosis. Two patients had elevated creatine phosphokinase (CPK; up to 642 and 1507 mcg/L) and elevated lactate dehydrogenase (LDH; up to 538 and 1261 IU/L); neither had myoglobinuria or evidence of renal compromise. No other post procedural laboratory abnormalities were observed. Adverse events are summarized in Table 3.

Post-procedural pain, occurring in 54% of DEE sessions, was managed with oral analgesics and usually resolved within two weeks. Skin toxicity, occurring in 42% of DEE sessions overall, was managed with topical treatment. No patient required surgical debridement or grafting for treatment-related skin injury.

Three cases (13%) in three different patients involved re-opening of previously healed wounds from biopsies, surgery, or in one case, a cryoablation procedure. This resolved with topical treatment and without infection in all cases. Transient regional alopecia was noted where the treatment zone included hairy skin.

No patient exhibited signs or symptoms associated with systemic doxorubicin toxicity, such as alopecia, myelosuppression, or nausea. One patient with a flank mass experienced a vascular spinal cord injury during cannulation of the 10th intercostal artery, from which the anterior spinal artery arose—neurologic symptoms improved from American Spinal Cord Injury Association (ASIA) stage B to D over three months.

## 4. Discussion

This report provides a combination of prospective and retrospective data regarding DEE in 24 patients. In this population of mainly treatment-refractory patients, DEE achieved PR or SD in most patients (87.5%) and substantial tumor volume reduction (by a median of 59%) accompanied by a loss of T2 MRI intensity (by a median of 36%). The procedure was safe and well tolerated.

Notably, DEE achieved response results (using RECIST 1.1) similar to those recently described for tyrosine kinase inhibitors [42,43]. Response using RECIST provides a useful comparison metric; however, its utility in DFs is limited, as DFs assessment is not well suited for two-dimensional measurements. DFs tumors are often large and almost always solitary; therefore, changes in volumetric measurements can be used as a more accurate assessment of response. The correlation between two-dimensional and volume measurements is even more complex when assessing non-spherical tumors; such was the case for one of the patients in this study, for whom a 65% decrease in volume corresponded to 10% decrease in RECIST-based greatest length (Appendix A).

The observed adverse events limited completion of therapy for 1 (4%) patient, which compares favorably with recent data regarding the use of systemic sorafenib or pazopanib to treat DFs, in which drug-related toxicities led to treatment discontinuation in 20% and 23% of patients, respectively [42,43]. Elevations in CPK and LDH associated with treatment of two large DFs were noted and likely reflect immediate tumor necrosis. Rhabdomyolysis has been described in a DFs patient upon initiating therapy with sorafenib [44] and as a more common complication of DFs cryoablation [45]. Our experience suggests that most patients could undergo complete treatment in 2–3 procedures, and that most extremity DFs can be treated in an outpatient setting. Patients undergoing DEE for large, intra-mesenteric, thoracic, or mediastinal tumors should be observed overnight.

The procedural risks of DEE, including the potential for neurovascular and visceral injury, warrants further discussion. It may be useful in this context to consider the difference between arterial targeting of DFs versus the more commonly performed trans-arterial treatment of liver malignancies. Intra-hepatic tumor embolization is facilitated by the intrinsic difference between tumors, which are entirely dependent upon arterial blood, versus hepatocytes, which are largely dependent on portal venous flow. The physiologic consequences of modest intra-hepatic non-target embolization are generally well tolerated and often inconsequential. In contrast, DFs and their adjacent viscera are equally dependent upon the same perfusing arteries. Consequentially, tumor targeting demands more vessel selectivity and often requires pre-treatment arterial consolidation with permanent (coils, micro-plugs, or glue) or temporary occlusive material and techniques. Often, DFs-adjacent anatomy is unforgiving of even minor non-target embolization. This is certainly the reality in treating DFs within the mesentery, mediastinum, pelvis, or near the CNS. The case-specific risks of non-target embolization must be considered for every patient, and in fact for every procedure, as arterial anatomy changes for a given patient after each DEE session.

Doxorubicin is a topoisomerase inhibitor, but this is unlikely to account for its efficacy in treating DFs, a disease of active, well-differentiated fibroblasts with a low mitotic index. Doxorubicin may affect DFs fibroblasts via iron-dependent free radical generation and p53 mediated apoptosis [46,47,48,49], as these have been implicated in the mechanism of doxorubicin-induced cardiomyopathy [23,47,50]. Coagulative necrosis and ischemia due to arterial embolization [29] could also contribute to DEE toxicity, although DEE delivers maximal dose using fewest particles, and no angiographic evidence of arterial stasis was observed. Future studies will explore such mechanistic and pharmacokinetic dynamics of DEE.

The study was limited by its prospective–retrospective design, which was necessary to obtain a large enough number of patients/sessions in such a rare disease. In addition, the study was limited by the lack of pharmacokinetic experiments to validate the hypothesis that doxorubicin concentrations are low in plasma and high in target tissues. Although this hypothesis was supported by prior studies [29], no pharmacokinetic data exist for DEE delivered outside the liver, where up to half of the non-metabolized drug may be excreted. Nevertheless, in the current study the bead-loaded doses were well below the recommended limits, and none of the systemic toxicities commonly associated with doxorubicin were reported.

Finally, we believe our results warrant consideration of selective arterial DEE in other soft tissue malignancies, especially those commonly treated with systemic doxorubicin. Pre-operative DEE may achieve faster and more robust target tumor responses than systemic neo-adjuvant treatment, while potentially facilitating resection by decreasing arterial perfusion.

## 5. Conclusions

Although limited by the small sample size and the combination of prospective and retrospective data, our findings suggest that DEE treatment is safe and achieves rapid clinical and volumetric responses in DFs.

## Figures and Tables

**Figure 1 cancers-14-05045-f001:**
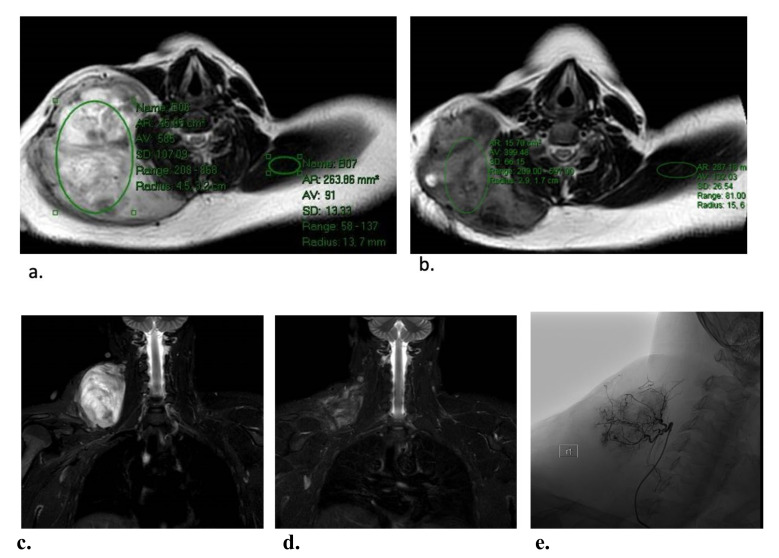
Representative images demonstrating T2 intensity reduction in a patient with a right base of neck tumor, after two DEE treatments. Semi-quantitative T2 intensity was determined by measuring the signal intensity within a representative area in the DFs relative to the signal of normal muscle on the same slice. The T2 intensity at baseline (**a**) was 585 and that in the contralateral trapezius muscle was 9.1 (i.e., intensity ratio of 6.4). The respective T2 intensities 2 months after two DEE treatments (**b**) were 399 and 132 (i.e., intensity ratio of 3.0). Thus, the overall reduction in T2 signal intensity was 47%. Corresponding coronal short tau inversion recovery (STIR) images from before (**c**) and after (**d**) therapy. The hypervascular tumor was supplied by a branch of the right ascending cervical artery (**e**).

**Figure 2 cancers-14-05045-f002:**
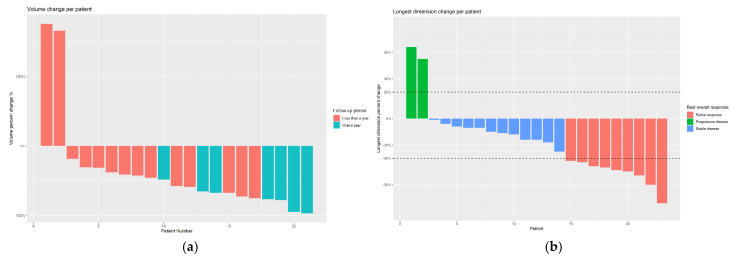
(a) Change in tumor volume per patient, where red and blue represent patients who were followed up for <12 months and ≥12 months, respectively; (b) change in the diameter of the tumor (longest dimension) per patient with corresponding RECIST 1.1 results (red, partial response; blue, stable disease; green, progressive disease).

**Table 1 cancers-14-05045-t001:** Patient/tumor characteristics and treatment summary.

Characteristic	Patients *n* = 24
Age, median (IQR), years	24 (16–34)
Sex, *n* (%)	
Female	15 (62%)
Male	9 (38%)
Tumor location, *n* (%)	
Chest/abdomen wall	7 (29%)
Neck/shoulder/axilla	7 (29%)
Mesenteric	3 (13%)
Mediastinum	1 (4%)
Extremity	6 (25%)
Tumor volume at baseline, median (IQR), mL	310 (108–686)
Tumor largest dimension at baseline, median (IQR), cm	10.5 (9.35–13.925)
Prior treatments, n (% of patients) ^1^	
None	7 (24%)
Surgery	8 (33%)
Systemic	16 (67%)
Other (cryoablation/isolated limb perfusion)	2 (8%)
Time from diagnosis to treatment, median (IQR), years	2.5 (2.0–3.6)
Number of DEE treatments, median (range)	2 (1–4)
Interval between treatments, median (range), months	2.3 (2–4)
Doxorubicin delivered per treatment, median (range) mg	49 (8–75)
Total doxorubicin delivered, median (range), mg	75 (8–269)

Abbreviations: DEE, doxorubicin eluting embolization; IQR, interquartile range. ^1^ A patient could have received more than one treatment.

**Table 2 cancers-14-05045-t002:** Efficacy results overall and by duration of follow-up.

	Median (IQR) Follow-up, Months	Median (IQR) Number of Procedures	Median (IQR) Reduction in the Longest Dimension	Median (IQR) Reduction in Tumor Volume	Median (IQR) Reduction in T2 Signal	Response (RECIST 1.1)
All patients (*n* = 23)	8 (4–14)	2 (1–3)	16% (7–36%)	59% (40–71%)	36% (19–55%)	PR: *n* = 9 (39%) SD: *n* = 12 (52%) PD: *n* = 2 (9%)
Patients with follow-up ≥12 months (*n* = 7)	33.2 (22.7–53.9)	2 (2–3)	37% (25–45%)	76.4% (66.2–86.3%)	56.7% (32.6–66.8%)	PR: *n* = 5 (71%) SD: *n* = 2 (29%) PD: *n* = 0
Patients with follow-up <12 months (*n* = 16)	5.2 (3.4–8.1)	2 (1–3)	10.5% (6–27%)	44% (31–62%)	33% (15–45%)	PR: *n* = 4 (25%) SD: *n* = 10 (63%) PD: *n* = 2 (13%)

Abbreviations: IQR, interquartile range; PD, progressive disease; PR, partial response; SD, stable disease.

**Table 3 cancers-14-05045-t003:** Safety summary (out of 52 DEE sessions).

Adverse Event	Any Grade, *n* (%)	Grade 3–4, *n* (%)
Post-embolization pain (per treatment)	28 (54%)	0
Skin injury	10 (42%)	0
Post-embolization pain/fatigue > 1 week (per treatment)	5 (21%)	0
Neuropathic pain	4 (17%)	0
Reopening of wounds	3 (13%)	0
Local alopecia	2 (8%)	0
Neurovascular injury	1 (4%)	1 (4%)

## Data Availability

The datasets used during the current study are available from the corresponding author upon reasonable request.

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
