# Peer review of "Selective Intra-Arterial Doxorubicin Eluting Microsphere Embolization for Desmoid Fibromatosis: A Combined Prospective and Retrospective Study"

_cancers, 2022, doi:10.3390/cancers14205045_

Round 1

Reviewer 1 Report

This is a very interesting and clinically significant article that reports the results of a clinical trial examining the efficacy of selective intra-arterial doxorubicin eluting microsphere embolization for desmoid tumors.

The abstract, introduction, materials and methods, and results are all very appropriately described.

As for the discussion, please consider the following.

#1. Unfortunately, neurovascular damage was observed as a serious adverse event in one case. I would like to see a more in-depth discussion on whether this case was an appropriate indication for this technique, and what kind of cases would be good to target for future dissemination of this technique.

#2. Is there any prospect to use this technique as preoperative adjuvant therapy for soft tissue sarcomas other than desmoid tumors? Or has it already been researched and developed? Please add your thoughts on the expansion of indications and development of this technique.

Author Response

Thank you for your review and thoughts on our manuscript. With regard to point #1 - thanks for this insightful question. We have elaborated on the required sensitivity to neurovascular and cns structures near DEE arterial supply in the manuscript. 

Regarding the use of DEE in pre-op adjuvent therapy for soft tissue sarcoma (point #2) we indeed have begun assessing this and, in addition, use the same approach for non-operative patients as palliation. This point also has been added to the discussion. Thanks.

Reviewer 2 Report

Manuscript ID: 1892754

Type of manuscript: article

Title:  Selective Intra-arterial Doxorubicin Eluting Microsphere 2 Embolization for Desmoid Fibromatosis: A Combined 3 Prospective and Retrospective Study

Journal: Cancer

In this article, the authors have investigated the arterial doxorubicin eluting embolization, an approach characterized by high drug concentrations in the tumor alongside limited systemic drug exposure The study included 24 patients.

The paper is well done and presented. All the criteria for a correct presentation were respected. The research was conducted on patients with informed consent and with the acceptance of ethical committee. The study included not too many, but a sufficient number of patients, it was performed for a relatively short period of time, but probably just sufficient. It is well described, and the results are conclusive and helpful.

I personally recommend the publication of this paper.

I have only a suggestion:

I suggest to describe better the hypothesis in the context of relevant research in the field, in the introduction

Author Response

Thank you for reviewing our manuscript and for your thoughtful suggestion. We have indeed expanded the discussion of the hypothesis (DEE as an option for Desmoids) to include more historical context (liver TACE/ Radioembolization) and more contemporary, expanded research into arterial drug delivery.  Highly appreciated!